# State Involvement in LGBT+ Health and Social Support Issues in Canada

**DOI:** 10.3390/ijerph17197314

**Published:** 2020-10-07

**Authors:** Nick J. Mulé

**Affiliations:** School of Social Work, Faculty of Liberal Arts and Professional Studies, York University, Toronto, ON M3J 1P3, Canada; nickmule@yorku.ca; Tel.: +1-416-736-2100

**Keywords:** Canada, health inequities, LGBT+, social support

## Abstract

For the first time, the broad health issues, needs and concerns of LGBT+ people in Canada were taken up by the federal government’s Standing Committee on Health in 2019. The findings of their consultations with LGBT+ Canadians produced a report that at once captures the breadth of input received, and provides an opportunity for accountable state response to LGBT+ health needs in the form of research, education, policy, funding and programming, yet questions arise as to the socio-political approach that will ultimately be taken. This focus on the health of LGBT+ Canadians follows decades of grassroots and sometimes state-funded research on this very issue. This study undertook a critical content analysis, premised on the queer liberation theory of *The Health of LGBTQIA2 Communities in Canada* report issued by the Standing Committee on Health. Although the report, for the most part, covers a breadth of broad LGBT+ health issues (a noted shift from the predominance of HIV/AIDS), the depth to which the Standing Committee took up and absorbed such issues is far less apparent. The heavy emphasis on entry-level recommendations by which to take up important LGBT+ health issues undermines a more progressive, liberationist approach that would more effectively address these concerns.

## 1. Introduction

In recent years, state and regional governments have shown an increased interest in matters relating to lesbian, gay, bisexual, transsexual, transgender, Two-Spirit, queer, intersex, etc. (LGBT+) health. This has followed years of LGBT+ community activism and community development towards addressing the health and social care needs of these communities. Preceding these events was the HIV/AIDS epidemic and concerns about breast cancer in lesbians. Much was learned from these health crises and utilized in drawing increased attention to the broad health and wellbeing concerns of LGBT+ populations. However, addressing LGBT+ health often requires state support, which, through its policy, programming and funding apparatuses leads to what Epstein [1] terms as an ‘inclusion–difference paradigm.’ This paradigm challenges LGBT+ communities with the need to be included within health and welfare safety nets that often come with constraints at the price of liberation.

There are a number of examples of regional and state governments that have begun to take up LGBT+ health issues with inquiries, consultations and reports. The European Commission conducted a two-year pilot project on the health of LGBTQ people between 2016 and 2018. The consortium that undertook this research included the International Lesbian and Gay Association (ILGA) Europe, and partners from Italy, the UK, Poland and EuroHealthNet [2]. This project involved research, developing and piloting a training package, hosting a conference and produced a series of outputs that included a review of the findings, focus group studies, a training course geared for healthcare professionals, evaluation reports and a conference summary report [3]. An inquiry about LGBT communities and health and social care was conducted by the UK Parliament’s Women and Equalities Committee [4] that was launched in 2018 and involved a survey of 108,000 LGBT people, 100 written submissions and testimony of 24 witnesses, before releasing their full report in 2019 [5]. In Canada, the House of Commons Standing Committee on Health undertook a study of the health of LGBTQIA2 communities in 2019. This study received 44 written briefs, heard from 33 witnesses and the Committee gathered information from health service organizations who address LGBTQIA2 health issues in four cities [6]. Although the US Department of Health and Human Services had produced annual reports on the health of LGBT Americans, its last report was issued in 2016 [7] and has not continued under the Trump administration.

Governmentality is a Foucauldian concept [8] that theorizes governmental surveillance and control of individuals or groups typically targeted for subjugation due to their non-normative sexuality [9], gender identity or expression [10]. In a broad sense, this form of governmentality speaks to omnipresent systems of power that govern subjects and is referred to as “the conduct of conducts” [11], (p. 11). Such panoptic governmentality plays out within neoliberal regimes of power focused on identity-based communities [12], with the healthcare system being one site of surveillance and control often on the basis of risk, making groups who engage in risk beyond norms, ‘hypervisible’ to neoliberal structures of regulation [13,14]. Cognizance is important regarding how the state via the healthcare system frames and operationalizes the health issues and concerns of LGBT+ people [1]. Going from repressive silence that renders certain types of sexuality invisible to the emergence of the LGBT+ movement, has moved these communities into hypervisibility, controlled by government surveillance in either realm [15]. 

How LGBT+ people are represented to the state comes from many sources including social scientists, LGBT+ social movements and individuals, yet noted tensions exist in India among and within these bodies [16]. From a larger structural perspective, even when LGBT+ recognition is achieved in law, such as the Employment Equality Framework Directive (EED) in the EU, implementation is problematic, particularly in countries for which social services are substantially provided by religious organizations [17]. Oftentimes, it is the equality-based frameworks such as ‘Equality and Diversity’ in policymaking and practice, as was taken up in UK local governments, that position LGBT+ people within homonormative standards [18]. Even in ‘progressive’ settings such as Canada, governmental recognition of LGBT+ people is layered, in that resistance to these populations persists [19], where a Prime Ministerial apology and expungement policies for past framings of gay sex as indecent serve homonationalist agendas, upholding political liberalism and homonormative family values [20], while sustaining homophobic public policy such as the unequal age of consent (legal age at which one can consent to sexual activity), resulting in the coexistence of legal homophobia and homonationalism [21]. In the health sector, state recognition can be at once emancipatory regarding the inclusion of LGBT+s in policy, yet restrictive, as seen in the governmentally monitored and controlled public health authorities’ directives on sexual practices [9]. 

Due to juridical frames for the inclusion of LGBT+ people in rights claims, as driven by LGBT+ social movements, the category of ‘sexual citizenship’ was created, one that primarily constitutes a liberal homonormative self, aligned with equality-based policies. Such restricted formulations inevitably exclude cultural, racialized and sex-positive policies, among others, in its Euro-American configuration, underscored by colonial and orientalist ideation [22]. Differential internal and external factors figured in the policy reinvention and diffusion of gay and transgender anti-discrimination laws [23], and the wide-spread growth of homonormativity through policy learning, a mechanism in which policymakers learn from each other, that assisted in the diffusion of same-sex marriage [24] across the US. However, the persistence of a unified voice in dialogue with broader societal perspectives and state policymaking is often anchored by a cultural basis that is premised on commonalities within LGBT+ political movements, yet also supports differences within [25]. The importance of internal differences in the conflated notion of LGBT+ in policy realms of the state can work against specified issues and interests of trans communities [26] and LGBT+ disability and intersectional communities [27]. 

To address the inclusion–difference paradigm in the recent interest of the state on LGBT+ health issues, I premise my critical analysis on queer liberation theory. Premised on the principles of celebrating differences, being open to the full spectrum of essentialism and fluidity and respecting the use of identities to further emancipatory social change, queer liberation theory effectively rejects assimilationism and homonormativity and sees serious limitations to rights-based and equality agendas [28]. Drawn from the historical principles and tenets of the gay liberation movement of the 1960s and 1970s [29], queer liberation theory is critical of the mainstreamed path the movement has taken [30]. Going beyond this historical backdrop, queer liberation theory promotes a progressive, sex-positive and a gender-diverse affirming queer politic that also encapsulates an intersectional analysis recognizing an array of social locations and assemblages [31], yet does not lose sight of gender and sexual diversity. This theory assists in separating out initiatives that lead to equality-based, homonormative notions of acceptance and respectability [32] and creative approaches that capture those marginalized from within [33]. The increased visibility of LGBT+s in society and more formally within the state also recognizes the state’s role in administrative violence [34], when attempting to address the needs of these communities. 

With growing governmental interest in the health of LGBT+ populations, this paper takes a close look at the Canadian government’s interest, how it sought information and input, and what it produced based on the feedback received from Canadian LGBT+ activists, community groups, health and social care workers and academics. This examination is based on *The Health of LGBTQIA2 Communities in Canada* [6] report and the contributions that were made to it. The content analysis examines the Report’s recommendations and, by turn, how the Canadian government chose to represent LGBT+ health issues. The engagement between the state and LGBT+ actors with health concerns is then deconstructed to carefully consider the implications of such relations on notions of inclusion, difference and liberation. This deconstruction, based on critical content analysis, has implications not only for Canada, but for the other states referenced, with a similar view on LGBT+ health, and those who have yet to do so. 

## 2. Methods

Critical discourse analysis [35] was employed to qualitatively examine *The Health of LGBTQIA2 Communities in Canada* [6] report and, in particular, its recommendations. By undertaking a critical discourse analytical review of this report, an LGBT+ socio-cultural and political lens was utilized to determine how LGBT+ health issues are understood, framed and communicated. The Standing Committee on Health based its report on submitted briefs, witness presentations and visits to LGBT+ health service providers [36] contributing to its content. In keeping with the ‘inclusion–difference paradigm’ and queer liberation theory, a deductive approach was undertaken that draws from the existing literature on health issues affecting LGBT+ communities and over two decades of the author’s LGBT+ health research and activism. This deduction produced the following themes: LGBT+-based health issues; equality-seeking measures; difference highlighted concerns. The results of the findings in each of these themes are then further discussed with regard to their implications when diverse LGBT+ perspectives on health engage with the identity-based, categorizing world of policymaking. 

This being the first federally commissioned report on the broad health issues of LGBT+ communities in Canada, there were no other similarly prepared Canadian reports to compare it to. This, in turn, speaks to the significance of this report. This is not to indicate that the report will necessarily have political influence, as this depends on whether the government in power will pay it any attention. However, using the means by which the government pursued the consultations via an open call and the report as its output, undertaking a critical discourse analysis of the report premised on queer liberation theory and applying an LGBT+ lens provides an examination of the report that is not consistent with the approach of the Standing Committee on Health or all who participated in the consultations. Hence, this critical discourse analysis provides a means of understanding the issues and how they are to be addressed from an alternative angle. This being said, a limitation of the study is that it focused solely on the report produced by the Standing Committee on Health to examine how they interpreted the input they received and the direction that they are recommending the federal government go in, in addressing LGBT+ health concerns. Future research could also examine the contributions the Standing Committee on Health received during the consultations, including the 44 submitted briefs received, 33 witness presentations given and notes on the LGBT+ health service providers visited in four cities. Such an examination may further reveal the diversity of perspectives and approaches to LGBT+ health from the activists, researchers and health care providers working in this area.

## 3. Results

### 3.1. The State and LGBT+ Health – Process of Engagement: Terminology

For the purposes of this paper, the state refers to the Canadian federal government and in particular the Standing Committee on Health of the House of Commons of Canada. Terminology is important, especially when attempting to engage with a marginalized community such as the gender and sexually diverse. The importance of terminology is based on both the choice of reference and how such communities are being represented. It is noted that the acronyms used to reference the gender and sexually diverse often change, have become expansive over time, and vary with regard to order, specificity and inclusion. From the outset, the state chose to use the awkward acronym LGBTQIA2 to refer to lesbian, gay, bisexual, trans, queer, intersex, asexual and Two-Spirit Canadians, closely resembling the LGBTQ2 Secretariat, which deals with LGBT+ issues at the federal level, in name, without rationale for the latter [37]. Although somewhat familiar at first glance, it is in the breakdown that the awkwardness reveals itself. Reducing the ‘t’ to trans only risks the conflation of transsexual and transgender individuals and their respective health issues. The inclusion of asexuals may be seen as controversial to some, who have not seen this group as part of the gender and sexually diverse, but this is arguable. The more specified disregard is the mere tagging on of ‘2’ to the acronym to represent Two-Spirit people, when they are usually acronymized as ‘2-S.’ The state undermines the importance of the Indigenous populations and those therein who experience and/or identify as being Two Spirited in terms of their gender identities and sexual desires. In fact, the report has a section on ‘Definitions and Terminology’, in which language is explained by some of the witnesses that came before the Standing Committee, yet the latter settled on LGBTQIA2, including it as part of the title of its report.

Further to the use of LGBTQIA2, when the report turns to an encompassing phrase, it opts for the dated ‘gender and sexual minorities,’ terminology that has not been used since the late 1980s and early 1990s. What is problematic about this terminology is that it emphasizes the quantitative, placing far less emphasis on the qualitative experiences of these populations or understanding of the fluidity of gender and sexual feelings and experiences that can broaden to numbers beyond those who identify as such. The report does capture the more currently used terminology of ‘gender and sexual diversity’ to entitle its glossary in one of its appendices, yet it is unknown as to why this was not adopted as the encompassing terminology throughout the report. Language has an important place in discourse and when the state is attempting to engage with a marginalized population on an important topic such as health care, developing, understanding and embracing terminology that speaks to the relevance of these often under-represented communities is crucial to the consultative process. Doing so communicates that these communities are being taken seriously while recognizing the evolution these communities are experiencing and how this is captured in phraseology.

### 3.2. The State and LGBT+ Health – The Responses

With a timeline of roughly a month to respond to a call for submissions on LGBTQIA2 Health in Canada in the spring of 2019, the Standing Committee on Health received 44 briefs. Additionally, the Committee heard from 33 invited witnesses from across the country. The Committee also traveled to meet with organizations that provide health services to LGBT+ communities in Montreal, Winnipeg, Calgary and Vancouver. Respondents to the call included academic researchers, community-based research organizations, provincial agencies and volunteer peer-support groups. The majority of these respondents specialize in gender and sexually diverse health issues, from research to services; others were more mainstream yet encompassed the gender and sexually diverse communities. The report’s 23 recommendations, which are sub-sectioned into 11 subjects [6], are now delved into as they apply to three themes deduced from the report.

### 3.3. LGBT+-Based Health Issues

This theme looks at the types of LGBT+ health issues that respondents chose to focus on, including whether they were illness-based or broad health and wellbeing-based and/or spotlighting particular segments of the LGBT+ population. Of importance with regard to this consultation was that it represented a shift—an opening—on the part of the state to engage in a dialogue regarding the broader health and wellbeing issues of Canada’s gender and sexually diverse following decades in which the HIV/AIDS pandemic dominated the health discourse of these communities. Such domination had direct implications on the research, funding, programming and policy making associated with most, if not all health concerns regarding these populations, resulting in an illness-based approach to the detriment of addressing broader health issues [38]. However, the continued presence of HIV/AIDS and its marked impact on gender and sexually diverse communities, despite the availability of anti-retrovirals and protective measures such as PrEP, not disregarding the accessibility issues for the latter which persist in Canada [39], warrant attention. Only two of the 44 briefs and three of the 33 witnesses that presented to the Committee, were from HIV/AIDS organizations, while two represented broader health issues of men who have sex with men (MSM), the latter representing a shift in missions from disease-based to a broader health and wellness focus.

The themes of recognition and consistency of inclusion continue through the five recommendations under ‘Sexually transmitted and blood-borne infections.’ LGBT+ people and their specified health needs have yet to be captured in the licensing of home test kits for HIV and other sexually transmitted and blood-borne infections; calling for the human papillomavirus vaccine to be universally covered; including prescription drug coverage of antiretroviral drugs; updating guidelines on STIs to include gender and sexually diverse communities with links between sexual health and mental health, and, finally, to increase funding for the Federal Initiative to Address HIV/AIDS to CAD 100 million, which was recommended 17 years ago [40]. The omission of LGBT+ populations in significant health aspects covered by these recommendations in and of themselves demonstrate the extent of exclusion these populations have experienced to the detriment of their health.

The majority of witnesses and briefs represented broader, more sweeping concerns of the mix of LGBT+ people and the varying health issues that they experience. Some were more specifically focused on varying socially located identities such as trans, Indigenous, and youth. Although the majority of respondents (both briefs and witnesses) represented gender and sexually diverse groups and organizations and/or were self-identified LGBT+ individuals, six organizations appeared as witnesses and 13 briefs came from mainstream groups and organizations whose work encompasses the health concerns of these communities. The variety of respondents that represented government bodies, health care and social service providers as charities and non-profit organizations, academics and community-based researchers, and LGBT+ advocacy groups and organizations is also of note. The breadth of respondents that participated in this consultation is remarkable, not only in providing input in the relatively short timeframe of the call (given the busyness of academics, activists, and health care workers), but more importantly, the number of groups, organizations and individuals, the majority of which are actively working to address the healthcare and wellbeing issues of Canada’s gender and sexually diverse. This speaks to the development of a specialized subset of LGBT+ health within the broader Canadian health care sector, despite minimal state support.

### 3.4. Equality-Seeking Measures

Examined in this theme is the angle of the pitch for LGBT+ health recognition, premised on inclusion based on an equality argument (i.e., if heterosexual and cisgender Canadians have certain health supports, so too should LGBT+ Canadians). At the outset, this is a commendable pursuit premised on the principle of equal recognition of citizenship. Anything short of this reveals contradictions in the inclusion of sexual orientation and gender identity and expression in human rights legislation in Canada [41]. However, equality has its limitations, particularly when attempting to address the specified needs of a minoritized population that differ from those of the majority. Inclusion based on equality will only elevate to existing heteronormative levels, whereas inclusion based on liberation [42] will centre and, more specifically, address LGBT+ health needs. The former is equality-based inclusion, which tends towards assimilation whereas the latter is equity-based inclusion, which tends towards liberation. 

Importantly, the recommendation under ‘Research funding’ calls for one of the Tri-Council agencies, the Canadian Institutes of Health Research (CIHR), to be included in its mandate on sexual orientation and gender identity, yet it falls short on a few fronts. The recommendation omits gender expression and targets, specifically the Institute of Gender and Health within the CIHR, failing to recognize the importance of acknowledging and including LGBT+ people in other institutes of the CIHR such as the Institutes of Aging, Health Services and Policy Research, Human Development, Child and Youth Health, Indigenous Peoples’ Health, Infection and Immunity, Neurosciences, Mental Health and Addiction, and Population and Public Health (see ‘Data collection’ above). Where this recommendation falls shortest is in not calling for the codifying of LGBT+ research as a recognized area of study, an omission across the Tri-Council agencies in Canada. Research findings can have a direct impact on ‘Program funding’, and its recommendation that a program of grants and contributions be established through Health Canada and the Public Health Agency of Canada for LGBTQIA2 health issues is a start. Shortcomings in producing data in the former can negatively impact the latter. The Report’s call to ‘Target LGBTQIA2 communities within existing public policies and programs’ via the formation of an advisory committee on sexual and gender minorities to support established departments on implementing LGBTQQIA2 community-specific measures in such areas as housing, homelessness, poverty reduction, tobacco, drugs and substance use speaks to an integrationist approach, with limiting effects on fundamental change. Many of these issues are currently being taken up as national strategies, but the federal government will not consider a similar nation-wide strategy for LGBT+ populations.

### 3.5. Difference Highlighted Concerns

As an alternative to the previous theme, the pitch here frames the health issues of LGBT+ people as distinct, premised on minoritized sexual orientation, gender identity/expression and/or sexual characteristics, in turn calling for specified health care responses. Although the Committee appeared to hear the call for a nuanced understanding of health issues affecting the LGBT+ communities in Canada, the report is framed by a dated equality/assimilationist agenda rather than a contemporary equity-based approach despite its claim to be looking at LGBTQIA2 health inequities. Most recommendations read as equality driven in terms of mere inclusion of, with less emphasis on specific health care responses to, LGBT+ people. However, the report does note the health specificities within the collective gender and sexually diverse communities, recognizing some health differences between groups (e.g., bisexual, trans health needs). The report also avoids pathologizing LGBT+ people, reporting on the contributing factors they heard from respondents that exacerbate health inequities with these populations such as discrimination, stigmatization and the effects of intersecting social locations.

Under the subject matter of ‘Awareness campaign, education and training’, a national awareness campaign is called for regarding the stigma and discrimination the gender and sexually diverse communities face, inclusive of an intersectional analysis, now decades after most provinces and territories have entrenched human rights legislation based on sexual orientation and, in the last decade, gender identity and expression. Other related recommendations include developing information tools in both official languages; inclusion of sexual orientation and gender identities and expressions in the *Canadian Guidelines for Sexual Health Education*; instituting cross-governmental (federal–provincial–territorial) relations in providing age-appropriate education to children, youth, parents and caregivers; and to have the federal government, in collaboration with the provinces, territories and provincial health professional and regulatory bodies, to formulate a working group to determine ways to promote education and training of health care professionals. Although all are commendable, they inevitably highlight the lack of attention paid to these communities to date. Numerous previous federally and provincially funded studies on the health needs of LGBT+ people produced similar, pointed and action-oriented recommendations over the last 25 years. A far more progressive and impactful approach would be to devise a national LGBT+ health strategy [43] with specified goals, objectives, timelines and evaluation components.

The recommendation under ‘Consultation’ is focused on trans and non-binary individuals, only regarding data collection on gender information and identification options for non-binary people. These are important issues to assist trans and non-binary Canadians in navigating the current system, yet they are incredibly limited, given the numerous other areas in which consultations could take place (surrogacy options for same-sex couples or LGBT+ individuals; gender non-conforming and/or trans children; addressing the needs of intersex infants; etc.). ‘Data collection’, another subject area, recommends that the federal government, through Statistics Canada (the national statistical office), consult with LGBT+ organizations, researchers and individuals for the development of questions for all its surveys on sexual behaviour and attraction; promote oversampling of LGBT+ populations to produce sufficiently sized samples for intersectional analyses; and include specified questions on its surveys regarding sex at birth, gender identity and sexual orientation regardless of the age of respondents, and that this be prioritized for health, housing, income, homelessness, as well as alcohol, tobacco and other substance use surveys. There is no mention of the Canada Census and its slant towards same-sex couples and their marital status and how this further marginalizes LGBT+ individuals not in coupled relationships [44]. Further, such recommendations overlook the work that LGBT+ groups have done over the years consulting with Statistics Canada for proper representation [45]. In other words, this has and continues to happen and, as such, the focus needs to be placed on implementing what has long been called for. 

The recommendations under ‘Health for trans people’ call for the inclusion of trans health issues such as coverage for hormone costs and uniform coverage of gender-affirming surgeries across the country. Both are considered crucial for these populations with regard to full recognition of their enumerated human rights in Canada [41] and for consistency with the principles of the Canada Health Act [46]. Some of the recommendations from the report speak to more current demands that LGBT+ activists and scholars have been calling for. These include a call for the elimination of conversion therapy, including modifications to the Criminal Code; *consider* postponement of genital normalizing surgeries on children until they can participate in the decision-making process with the exception of a delay risking the child’s health; and a call to end all discriminatory practices with regard to blood, organ and tissue donations with regard to men who have sex with men and trans people. Although current, each of these issues have been challenged by LGBT+ activists and such recommendations called upon by scholars [47,48,49], once again demonstrating the slow uptake of the federal government on these issues.

## 4. Discussion

A major policy conundrum the report falls into is that of identification and categorization to fit the LGBT+ communities into the mainstream of society or capture the mainstream within these communities. The report’s equality-driven approach does not address these concerns. There is also historical amnesia in that nowhere in the report is it mentioned that the federal government funded a number of community-based studies on the health issues of LGBT+ people, beginning as far back as the late 1980s through the early-to-mid 2000s, each producing numerous recommendations [50], none of which were taken up in any formal way during that time. Apart from the shortcomings of existing data sources, such as Statistics Canada’s Canadian Community Health Survey (CCHS) (although it is inclusive of sexual orientation and gender identity/expression, it is limited on age range, sexual behaviour and sexual attraction and intersectionality with other identities and experiences), the impact on future research is not addressed, such as the Tri-Council being devoid of any recognition of LGBT+ populations as a specified area of research. Yet, as recommended by many of the respondents, the Committee acknowledges the importance of having direct involvement with LGBT+ communities in policy and program development and any decisions affecting their health.

Although on the one hand the federal Government of Canada can be commended for taking up important issues regarding the broad health and wellbeing of its LGBT+ citizens, on the other, it has taken 23 years since the start of antiretroviral therapies [51] to shift their gaze beyond HIV/AIDS. Nevertheless, this commendation comes with a qualifier regarding the state’s attention to LGBT+ communities in Canada. As has become apparent by the Standing Committee on Health’s report [6] and, specifically, its recommendations, its focus is on bringing LGBT+ Canadians into its fold, with all its state-based governmentality. As discussed in the results section, most of the recommendations call for the inclusion of LGBT+ people in pre-existing heteronormative structures with a particular focus on equality. Where the report’s recommendations do focus on diversity, it tends to be primarily on trans and intersex issues. This emphasis on the pre-existing may make practical sense politically (it increases chances the of the recommendations being implemented within existing structures), but it restricts broader, liberationist means of actually creating new LGBT+ foci, such as an LGBT+ health strategy [43] recognizing these populations in the social determinants of health [52] or codifying LGBT+ studies as an area of research. 

The development of new structures or, at minimum, new components that recognize LGBT+ health issues within existing structures, would inevitably cost more financially. It would also cost more socially, as it would force the state to acknowledge at a deeper level the diversity that exists within the LGBT+ communities and how distanced some of those diversities are from Canada’s mainstream populations. The crucial step of actually addressing specific LGBT+ health issues structurally and systemically calls for a higher investment, which the report hardly signals. However, the cost of not addressing LGBT+ health issues effectively can be much higher due to pre-mature deaths caused by homophobia in areas such as smoking, substance use including alcohol and drugs, and suicide [53], not to mention the numerous social determinants of health disparities that lead up to them [52]. Holding such consultations and producing a report is one thing, what has been produced and what impact it will have is quite another.

Of concern is the direction the state will take in its governmental role and relationship with Canadian LGBT+ citizens [54] in addressing the latter’s health issues. This is not to say that should the recommendations of this report be implemented, the recognition would fail to advance the health issues of LGBT+ people, but rather that the state reifies what it knows and leaves little to no room for creative, innovative approaches designed by and for LGBT+ people rather than them having to reshape themselves to have to fit into pre-existing structures that were not originally designed for their purposes. This is an example of the inclusion–difference paradigm [1,55] where the state is aligning with the ‘inclusion’ component and not recognizing the ‘difference’ component. The question of the implementation of the report’s recommendations may be a moot point, for at press time, the federal government has not formally responded since the report’s release in June of 2019 [56]. Politically, the timing of this release took place as the country was heading into a federal election, with speculation that the Trudeau Liberal government may not survive it, leaving the status of the report in doubt. Post the 2019 federal election, the Liberals were returned to power albeit without a majority, then the COVID-19 pandemic strongly impacted Canada in March of 2020, with the focus having to shift. However, many LGBT+ people have faced further disparities due to the pandemic [57], and neither the report nor LGBT+ health issues in general are featured on the LGBTQ2 Secretariat website [58]. 

Another area of the report needing further attention is that of intersectionality. This topic was given a section in the report and highlights information the panel received from Health Canada and the Public Health Agency of Canada indicating they now undertake sex and gender-based analysis plus (SGBA+), which incorporates gender-diverse people, in program, policy and research development. Although a very important and positive development, broader issues of intersectionality regarding age, (dis)Ability, ethnicity, race, and socio-economic status among others, and how they intersect within the LGBT+ communities, are prominent concerns for these communities [59,60], yet are not reflected to this extent in the report. Given that the report calls upon the federal government to work with Canada’s LGBT+ communities to address the latter’s health concerns, during this extended time of inattention (not just due to the pandemic), discussions among LGBT+ academics, activists, health workers and their allies are important to ensure that the recognition and inclusion of diversified LGBT+ Canadians not be restricted to pre-existing structures only, and that equality measures not be the goal, despite use of the term ‘inequities’ in the report, for these will not further the diverse health needs of these communities. 

## 5. Conclusions

With governmental bodies such as the European Commission, the UK Parliament, the Government of Canada, and in the past, the US Department of Human Services, all taking an interest in the health issues of LGBT+ populations, it is incumbent upon LGBT+ communities to critically review how these issues are being taken up. State relations with LGBT+ communities often align agendas with mainstream heteronormative perspectives, which do not necessarily capture more marginalized members of LGBT+ communities. The recommendations put forth by the Standing Committee on Health for the Canadian House of Commons, for the most part, call for the inclusion of LGBT+ health issues being taken up in existing structures that are not designed to accommodate the complex diversity of these communities. At risk is the reification of these communities to more closely match the status quo while further marginalizing the health issues of the most vulnerable therein. More emphasis on the ‘difference’ component of the inclusion–difference paradigm will allow LGBT+ communities to approach their health concerns from a liberationist approach that creates new policies, funding and programming to meet their diversified health needs.

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
