# Peer review of "State Involvement in LGBT+ Health and Social Support Issues in Canada"

_ijerph, 2020, doi:10.3390/ijerph17197314_

Round 1

Reviewer 1 Report

The manuscript provides a discourse analysis of a Canadian federal report on LGBT health issues. While the intent of the report is largely commended, the author argues that owing to the discursive structure and consultation oversights/failures, that the report and its recommendations ultimately fall short of a liberationist or equity potential. In my view the manuscript is weakest in its early analysis and assertions (i.e., 3.1) and strongest in its latter analysis and assertions (i.e., 3.4, 3.5). The discussion is builds appropriately on the analysis. The manuscript appears properly referenced and draws on a range of appropriate academic and gray literature. Overall, the manuscript makes a convincing argument within the context of theory and the confines of the selected data. My main criticisms are directed at the introduction and the methods sections.

Introduction (Line 66-77): This content seems misplaced. The content unexpectedly jumps into governmentality but then doesn’t return to it again until the discussion, and even then only brief mention in one paragraph. Concepts like panoptic and repressive silence require a bit attention if they are to be broached. My concern with this paragraph is that it reads a bit jargon-heavy for this journal. Given that the manuscript seeks to center the queer liberation theory angle, perhaps the place to engage with the governmentality point is solely in the discussion.

Methods (139-149): The methods discussion is inadequate and needs to be extended. The manuscript inconsistently refers to the methodology. I’m familiar with content analysis, discourse analysis, and critical discourse analysis, all of which are distinct approaches with the latter two most closely related. I’m sure readers from other disciplines have different understandings also. The specific methodology that was used needs to be identified and explained. Further, the deductive approach needs to be explained. Deduced from what? A pre-read of the report and if so which section(s), previous research, the author’s experience, the literature, other? A bit more information on why the report was selected would also be helpful. Why choose this document, how/why does matter at this time, what sets this document apart from others past and present? Some of this detail is alluded to in the intro but more detail is needed so that readers comprehend that the report isn’t some esoteric document that lacks any real clout. Finally, some discussion of limitations of the study should be present either in the methods or the end of the discussion.

Minor points/typos

Line 24: “shifted” = shift

Line 66: “Foucauld” = Foucault

Line 87: “persist” = persists

Line 119: missing word. Recommend either inserting “a” before “gender” or making politic plural?

Line 168-170 (else too perhaps): recommend “two-spirit” instead of “2-spirit” except when directly quoting the report.

Line 240: The analysis notes that a month is a “relatively short timeframe” for the committee to receive stakeholder input. Relatively short compared to what? Is this a short timeframe? It’s a relatively short report. Is it your sense that this was rushed, or that other committees provide more time for input?

Line 259-262: Awkward sentence. Recommend comma and word insertion: “Importantly, the recommendation under ‘Research funding’ calls for one of the Tri-Council agencies, the Canadian Institutes of Health Research (CIHR), to include in its mandate sexual orientation and gender identity, yet it falls short on a few fronts.”

The analysis gets a bit confusing at Line 285. The sentence seems to critique the report for using the language of equity while pursuing equality-based recommendations. My understanding of the manuscript’s argument is that one of the main problems with the Report is that it pursues a dated equality/assimilationist agenda rather than a contemporary equity-based approach. Is there some word slippage in the sentence or is another point being made by the analysis?

Line 297-303: Sentence gets long and difficult to follow. Recommend starting new sentence with “Although all commendable…”

Reviewer 2 Report

The article deals with a topical issue of interest that should be made visible. However, the author's approach is unrithe and objective, and this loses scientific quality to the article.

The theoretical introduction is well-founded, but, as it is an article that analyzes a local situation, the article should be further detailed at the local level and compared to other health systems so that the reader can be placed. The author should bring together local and global levels in a more appropriate way.

Methodology is described with very little scientific rigurosity. The article names the method to be used without indicating the possible peculiarities of this study.

Similarly, the results lack objectivity. Even if it is a critical analysis, opinions should be somewhat more informed.

The discussion repeats many of the ideas shown in the results, without providing relevant information or strengthening the results with supporting bibliography.

At the format level, the bibliography must be reviewed to strictly comply with subpoena rules.

Reviewer 3 Report

It is a very interesting paper and it provides new ideas for reducing health inequities based on sexual orientation.

In this manuscript, the author examines the Canadian government’s interest in the health of LGBT+ populations and how it sought information, input and what it produced based on the feedback from different LGBT+ actors. Then, the engagement between the government and the LGBT+ actors is deconstructed to carefully consider the implications of such relations on notions of inclusion, difference and liberation.  

The data source and measures both seem appropriate and the paper addresses an interesting and important topic. However, I have some concerns about some aspects of the manuscript that I outline below.

REVISION

  1. INTRODUCTION
    1. The first sentence of second paragraph (this sentence just include state) is practically the same sentence as the first sentence of first paragraph, maybe it would be more appropriate to use a different sentence to introduce the interest of state governments in LGBT+ health
    2. For people not living in Canada is complicated to understand the following sentence: the unequal age of consent resulting in the coexistence of legal homophobia and homonationalism (Smith, 2020).” What type of consent does the author refer to when talking about the unequal age of consent? There are different types of consent…
  2. METHODS
    1. More information would be advisable in relation to critical content analysis, it would be useful especially for those readers who don’t know understand the qualitative methodology
  3. RESULTS
    1. It is not necessary to use two adverbs (still and yet) in Page 5 Line 222-223: LGBT+ people and their specified health needs are still yet to be captured in the licensing
    2. There is a writing error in Page 7 Line 286 (I think the author wanted to use more instead of mere): as more equality driven in terms of mere inclusion with less emphasis on specific health care responses.
